# Immobilization of Caraway Essential Oil in a Polypropylene Matrix for Antimicrobial Modification of a Polymeric Surface

**DOI:** 10.3390/polym13060906

**Published:** 2021-03-16

**Authors:** Monika Strasakova, Martina Pummerova, Kateryna Filatova, Vladimir Sedlarik

**Affiliations:** Centre of Polymer Systems, University Institute, Tomas Bata University in Zlin, Trida Tomase Bati 5678, 760 01 Zlin, Czech Republic; strasakova@utb.cz (M.S.); filatova@utb.cz (K.F.); sedlarik@utb.cz (V.S.)

**Keywords:** caraway, essential oil, polypropylene (PP), talc, composite, antimicrobial packaging, thermoplastic processing

## Abstract

This study investigates antibacterial polymer composites based on polypropylene as modified by caraway essential oil at various concentrations, the latter immobilized on a talc. The caraway essential oil is incorporated in the polypropylene by a thermoplastic processing method. Analysis of the morphology of the composites was carried out by scanning electron microscopy. The chemical composition of the caraway essential oil in addition to its efficiency of incorporation and release were evaluated by GC/MS and Pyrolysis-GC/MS techniques, respectively. Determination was made as to the influence of such incorporation on thermal and tensile properties of the samples, while antibacterial activity was evaluated through conducting disk diffusion tests and measurement with adherence to the ISO 22196:2011 standard. It was found that incorporating the caraway essential oil in the samples did not affect the homogeneity of the thermoplastic-processed composites at any studied concentration. Stress–strain analysis confirmed the plasticizing effect of the essential oil in the polypropylene matrix, in addition to which, differential scanning calorimetry (DSC) and thermogravimetric analysis (TGA) analysis revealed that the prepared compositions with essential oil exhibited similar thermal properties to neat polypropylene. Results indicated significant antibacterial activity against *Staphylococcus aureus* and *Escherichia coli* at the concentration of essential oil of 4.9 ± 0.2 wt% and higher.

## 1. Introduction

About half the population of the European Union lives in flats [1], dwellings usually unsuited to storing fresh produce such as fruits and vegetables due to prevailing conditions. In this context, the potential occurrence of microbial pollution is an important matter in the safety of food and its points of origin, although the same issue affects cosmetics, hygienic materials, and other products. A host of diseases are associated with ingestion or use of a spoiled item. A high number of food-borne illnesses and related outbreaks have been reported in recent decades, pertaining to the contamination of fresh produce by polluted sources containing pathogenic bacteria, fungi, viruses, and protozoa [2]. Such problems have led to promotion of healthier lifestyles, in turn causing a rise in the consumption of fresh produce [3]. The scale of public demand for increased quantities of high-quality raw produce stemming from primary production can only be achieved by intensifying efforts in agriculture, ensuring effective pest control, and optimizing storage conditions to avoid spoilage of stored goods by germination or putrefaction [4].

As a consequence, there has been a move away from traditional packaging materials, void of any interaction with their contents, towards new concepts of active packaging with such interaction and synergy between the material utilized and the environment, with the aim of heightening quality and extending the shelf-life of goods [5].

Since public awareness of ecological matters has increased, demand by consumers for natural antimicrobial and antioxidant agents has been on the rise, with a shift away from synthetic preservatives [6]. To this end, attention has focused on essential oils (EOs) as potential bioactive additives, owing to their great inherent safety and lesser effect on the environment compared to metal or metal oxide nanoparticles (e.g., zinc, silver, copper and iron oxides), the latter being typically employed as antimicrobial additives in plastic materials [7,8].

EOs derived from plants are secondary metabolites that act as defence mechanisms for the same and its fruits and seeds. This has resulted in application of them for their medicinal properties, including antimicrobial and antioxidant activity, brought about by the given content of proteins, vitamins, and volatile compounds [9]. Due to its smell, flavour, and natural antimicrobial characteristics, EO derived from caraway is primarily utilized in the food industry for preserving goods and lending them an aroma [10]. EO sourced from caraway seeds is rich in γ-Terpinene, *p*-Cymene, pinene, cuminaledehyde, limonene, and carvone and boasts antimicrobial and antioxidant properties [11,12]. A major component is carvone, and its metabolites comprise monoterpenes in the derived EO that are of interest due to their antibacterial, antifungal, anticonvulsant, and cytotoxic qualities, for instance; the latter relates to cancer cell lines [10]. Since carvone is a major constituent of caraway EO, it is employed as a natural inhibitor of germination, especially for potatoes and onions in storage [13,14]. Applying liquid EOs on an industrial scale is difficult due to their inherent volatility, a problem overcome by encapsulating or incorporating the oil in a matrix.

An issue debated in society concerns what happens to polymeric materials at the end of their useful life for reasons of environmental safety and sustainability, as most packaging materials are non-biodegradable. Biodegradable polymers represent an alternative to conventional plastics, although certain limitations exist in extending their real-world application and furthering industrial utilization, such as high cost, insufficient mechanical performance, and inferior thermal stability [15]. A shift towards a circular economy is taking place, however, with the intention of increasing sustainability. Although forms of conventional plastics are still widely employed by manufacturers, an endeavour is being made to use recycled materials and heighten the utility value of the original ones. Plastics such as polyolefins can contribute to a circular economy, as they are recyclable and/or reusable [16]. Polypropylene (PP) is an example of one with numerous industrial applications as a consequence of its low cost, excellent chemical resistance, low density, and high tensile strength. Short glass fibres and talc are common fillers in PP systems [17]. The latter of the two is a hydrous magnesium silicate with the chemical composition of H_2_Mg_3_O_12_Si_4_, and it has long been utilized in cosmetics, polymers, roofing materials, paper, paint, and plastics [18,19].

Research in the literature describes the application of chitosan nanoparticles containing carvacrol or the EO of caraway as a means for extending the shelf life of goods, investigated by an emulsion technique without thermal processing [11,20]. Other studies discuss modification of a polymer by plasma treatment to bring about surface-level distribution of EOs [21,22]. Llana-Ruiz-Cabello et al. [23] characterized the antimicrobial activity of PP films supplemented with oregano EO and extract of allium for the packaging of ham by directly mixing the PP and the EOs with a thermal degradation temperature of ca 200 °C. Another focused on blown films of PP with the addition of 0–5% talc, which, following thermal processes, were impregnated after preparation with an EO extracted from lemons [24]. Such explorations investigate neither the mutual thermoplastic processing of polymer, filler, and EO nor any associated antimicrobial properties and release kinetics.

There is still a lack of data on thermoplastic incorporation of caraway EO into a PP matrix and the manner of its interaction with polymer composites. It is important to highlight that EO-based antibacterial agents are not easy to incorporate into polymer matrices due to their volatility. This drawback is overcome by immobilizing the antibacterial agents on inert carriers commonly utilized as plastic fillers. Herein, a study was made on the immobilization of caraway in a PP polymer matrix, aided by an inert carrier of talc for antimicrobial modification of the polymeric surface. The effect of incorporating the caraway essential oil was observed on the resultant thermal and mechanical properties, such as tensile strength, strain at break, and Young’s modulus. The antimicrobial activity of the EO was also evaluated against selected pathogenic microorganisms.

## 2. Materials and Methods

### 2.1. Materials

Caraway essential oil (EO) derived from *Carum carvi* L. was used as the antimicrobial agent. Its safety data sheet determined that it consists of four main substances (carvone, limonene, carvacrol, and myrcene); it was obtained from Prvni Jilovska (Jilove u Prahy, Czech Republic). Granules of homopolymer polypropylene (PP) of standard molecular weight distribution were sourced from SIBUR International GmbH (H032 TF/2, Vienna, Austria), the inert carrier talc (TC) of Ph. Eur. quality (CAS 14807-96-6) was supplied by IPL Lukes (Uhersky Brod, Czech Republic), and *n*-hexane (99%, CAS 110-54-3) was purchased from Penta Chemicals (Praha, Czech Republic). The media required for the microbiological studies—nutrient broth, soybean casein digest broth with lecithin and Tween 80 (SCDLP), plate count agar (PCA), and Mueller-Hinton agar (MHA)—were bought from HiMedia Laboratories Pvt. Ltd. (Mumbai, India). The bacterial strains of *Staphylococcus aureus* (CCM 4516) and *Escherichia coli* (CCM 4517) were obtained from the Czech Collection of Microorganisms, Masaryk University (Brno, Czech Republic).

### 2.2. Preparation of Samples

The TC carrier was activated under isothermal conditions at 160 °C for 30 min. Then, the EO was immobilized in the carrier by an isothermal process (25 °C) in a closed glass flask, which contained the caraway EO and the carrier at a ratio 1:2 (*w*/*w*) in order to obtain an additive suitable for thermoplastic processing. The PP masterbatch was prepared by a thermoplastic method with the EO incorporated on the TC in the manner mentioned above so as to achieve a theoretical content of EO at 20 wt%, TC at 40 wt%, and PP at 40 wt%. Thereafter, samples intended for testing were prepared from the masterbatch, neat TC and PP, as follows.

The mixture of the PP and the additive underwent melt compounding in an internal Brabender mixer (Plastograph^®^ EC plus, Mixer 50EHT32, Duisburg, Germany) equipped with two stainless steel screws and a bypass valve to allow continuous recirculation of the material. The neat TC and PP were heated to and melted at 180 °C, the mixer running for approximately 15 min at the operating speed of 110 rpm. After stabilizing the degree of torque, a calculated amount of masterbatch was added, the process continuing for another 5 min to achieve a sample with a defined theoretical content of EO. The designation and the composition of the samples are summarized in Table 1. Finally, the resultant products underwent compression moulding at 180 °C for 5 min in a manual press to create several films (each approximately 200 µm thick), which were subsequently cooled under the pressure of 10 MPa for 5 min.

### 2.3. Morphological Studies

#### 2.3.1. Scanning Electron Microscopy

A Phenom Pro unit (Phenom-World BV, Eindhoven, The Netherlands) was used to carry out scanning electron microscopy (SEM) at an electron accelerating voltage of 5 kV. Analysis was performed of the cryo-fractured parts of the neat PP film and composites in order to evaluate the degree of homogeneity and gain insight into the internal structure of the composites.

#### 2.3.2. Tensile Properties

The ASTM D882-12 [25] standard method was applied to gauge the tensile properties of the films on a tensile testing unit (M350-5CT, Testometric Co. Ltd., Rochdale, UK). Samples were prepared from strips of the films, with dimensions of 80 mm in length, 4.20 mm in width, and ca 200–220 µm thick. Prior to testing, all the samples were conditioned for 48 h at 23 ± 2 °C and relative humidity 50 ± 2% The samples were subsequently stretched at a speed of 50 mm/min, and the parameters of tensile strength, strain at break, and Young’s modulus were determined. Measurements were performed on five duplicates.

### 2.4. Thermal Properties

#### 2.4.1. Thermogravimetric Analysis

Thermogravimetric analysis (TGA) was carried out on TGA Q500 device (TA Instruments, Wilmington, DE, USA). Samples with an approximate weight of 10 mg were placed in the platinum pan and heated, after equilibration, at temperatures ranging between 25–800 °C at a heating rate of 10 °C/min in a nitrogen atmosphere at the constant flow rate of 60 mL/min. A data set was calculated from the percentage of additive remaining in each film after processing.

#### 2.4.2. Differential Scanning Calorimetry

The melting behaviour and the crystallization of the composites were studied by differential scanning calorimetry (DSC) on a differential scanning calorimeter (DSC 1 STARe System, Mettler Toledo, Columbus, OH, USA). Samples of the films at the approximate weight of 5 mg each were placed in aluminium pans. Nitrogen flow was set to 50 mL/min., and the following heating program was applied: an initial heating cycle from 25 °C to 200 °C (10 °C/min), maintaining the same for 2 min, and subsequent cooling to −50 °C (20 °C/min); this temperature was maintained for 2 min prior to conducting another heating scan to 200 °C. The melting temperature (T_m_) and the exothermal response relating to cold crystallization (T_c_) were obtained from the second heating cycle. The degree of crystallinity (*X_C_*) was calculated from the value for specific enthalpy of melting (Δ*H_f_*), as determined from the area under the melting peak in the second scan, applying the theoretical enthalpy of melting for 100% crystalline PP (209 J/g); see Equation (1) below [26,27,28].
(1)$$X_{C}\left(\%\right)=\frac{\Delta H_{f}}{\Delta H{{}^{\circ}}_{f}\times W}\times 100,$$
where Δ*H°_f_* is the theoretical enthalpy of melting for 100% crystalline PP, and *W* is the weight fraction of PP in the composite.

### 2.5. Qualitative Analysis of the Caraway EO

Stock solution of the essential oil derived from *Carum carvi* L. in *n*-hexane (4 µL/mL) was prepared for qualitative analysis by gas chromatography mass spectrometry (GC/MS). For semi-quantitative evaluation of its core components, carvone was utilized by means of a calibration curve at the range of concentrations of 4.8–480 ng/mL.

GC/MS was performed on a GCMS-QP2010 Ultra device (Shimadzu, Kyoto, Japan) equipped with a fused silica capillary column (Rxi-5ms, 30 m × 0.25 mm × 0.25 µm, Restek, Bellefonte, PA, USA). Helium was used as the carrier gas at a flow rate of 0.96 mL/min. The temperature of the injection was maintained at 230 °C at a split ratio of 1:50, the volume of the injected sample equalling 1 µL The temperature of the column was held at 50 °C for 1 min and then increased to 250 °C at a rate of 15 °C/min, a temperature which was maintained for 1 min. The temperatures of the ion source (EI, 70 eV) and the interface were set at 220 °C and 230 °C, respectively. The range of the scan was 35–350 (*m*/*z*) at speed 1666, and the entire programme lasted 15.3 min in total. The peaks obtained in the resulting TIC spectrum were identified with the help of the NIST11 Spectra Library.

### 2.6. Release Kinetics of the Caraway EO

The thermal desorption technique was employed to determine the release test, in addition to which the concentration assessment of the EO directly after thermoplastic processing was carried out. Analyses were conducted with the aid of a Multi-Shot Pyrolyzer unit, model EGA/PY-3030D (Frontier Laboratories Ltd., Fukushima, Japan), connected to the GC/MS device mentioned above. Samples of approximately 3 mg with different amounts of the caraway EO were weighed, and a part of each sample was removed for analysis (to gauge actual EO content following preparation of the samples; see Table 1). The remaining portions of them were heated at the temperature of 40 °C for a predetermined time for the release test. They were transferred into a pyrolysis cup and pyrolyzed at 250 °C for a duration of 1 min, which was sufficient for the sample to melt and ensure pyrolytic decomposition of its components did not occur. Separation was carried out on an Ultra alloy-PY2 capillary column (30 m × 0.25 mm × 0.5 µm, Frontier Laboratories Ltd., Fukushima, Japan). The temperature of 300 °C was applied for the GC injector and the interface between the pyroprobe and GC device; a split injector ratio of 1:150. The oven temperature of the GC equalled 80 °C and was held for 1 min, followed by two instances of continuous heating, initially (at 20 °C/min) to 170 °C and again (at 35 °C/min) to 300 °C for 4 min. Helium constituted the carrier gas at the column flow of 1.2 mL/min. The temperatures of the ion source and the MS interface equalled 250 °C and 300 °C, respectively. The scan speed was set at 1666 over a range of 35–350 *m*/*z*. Analysis was performed three times for each film sample.

### 2.7. Study of Antimicrobial Properties

#### 2.7.1. Disk Diffusion Method

The diffusion of the neat caraway EO, its solutions (10, 1.0, and 0.1 vol%), and EO from the films was determined by the disk diffusion method, according to EUCAST (European Committee on Antimicrobial Susceptibility Testing) standards [29]. *Staphylococcus aureus* CCM 4516 and *Escherichia coli* CCM 4517 were utilized for this purpose in bacterial suspensions of concentration 10^6^–10^7^ CFU/mL. The neat caraway EO or the solution with concentrations of 0.1% or 1.0% (in ethanol) of the EO were pipetted into wells (8 mm in diameter) on plates with Mueller-Hinton agar inoculated with the bacterial suspension (2 wells in a single Petri dish, with plates in duplicate; at the volume of 0.1 mL), followed by incubation at 35 ± 2 °C for 24 h. The films were tested in the same manner, the only difference being that disc-shaped pieces of the films (8 mm in diameter) were placed directly on such agar plates. Thereafter, the width of the inhibition zone for each sample was measured. The lack of effect by the ethanol solvent was confirmed by determining the zero zone of inhibition of pure ethanol against both bacterial strains.

#### 2.7.2. Measurement of Antibacterial Activity on the Plastic Surfaces

Antibacterial activity was tested according to ISO 22196:2011 [30]. Prior to this, the samples were disinfected with 70% ethanol in water. Two bacterial strains suspensions were utilized (*E. coli* 2.7 × 10^6^ CFU/mL; *S. aureus* 6.6 × 10^6^ CFU/mL), which were prepared in 1/500 nutrient broth. The subsequent bacterial suspension was applied on the surface of the samples (of dimensions 25 × 25 mm) at the volume of 100 µL; then, each sample was covered with inert polypropylene foil (of dimensions 20 × 20 mm). These were cultivated at 35 ± 1 °C and min. 90% RH for 24 h. After the period of incubation, the cover foil was removed, and each sample was completely washed with SCDLP broth. The pour plate method was employed to determine the viable bacteria count. Incubation at 35 ± 1 °C took place for another 48 h, at the end of which the number of viable bacteria recovered per cm^2^ per test specimen was enumerated in accordance with Equation (2):(2)$$\;N=\frac{100\times C\times D\times V}{A},$$
where *N* is the number of viable bacteria recovered per cm^2^ per test specimen, *C* is the average plate count for the duplicate plates, *D* is the dilution factor for the plates counted, *V* is the volume (mL) of SCDLP added to the specimen, and *A* is the surface area (mm^2^) of the cover foil.

The effectiveness of the plastic films expressed as antibacterial activity was calculated according to this Equation (3):(3)$$R=U_{t}-A_{t},$$
where *R* is antibacterial activity, *U_t_* is the average of the common logarithm of the number of viable bacteria (CFU/cm^2^) recovered from the untreated sample (blank), and *A_t_* represents the average of the common logarithm of the number of viable bacteria (cells/cm^2^), recovered from the treated test specimens after 24 h.

## 3. Results and Discussion

Antimicrobial blends for the preparation of films were created by the melt compounding technique, whereby commercially available caraway essential oil (EO) into the solid carrier of talc (TC) with polypropylene (PP) was mixed. The EO was added into the solid carrier prior to thermoplastic compounding with the polymer matrix, the purpose being to enhance thermal stability and reduce the diffusion rate for EO from the matrix. The subsequent films, after they had undergone compression moulding, were approximately 200 µm thick and light grey-green in shade, as typical for the given filler, with the aroma of caraway.

The effects of incorporating the EO into the composites on the properties of the resultant films were investigated by carrying out morphological studies, tensile tests, thermal analysis (TGA and DSC), gas chromatography, and antimicrobial evaluation.

### 3.1. Morphological Studies

#### 3.1.1. Scanning Electron Microscopy

It is known that a matter of importance regarding particulate composites is the interface between the filler and the matrix in addition to their distribution in, dispersion in, and interfacial adhesion with a polymer matrix [31,32]. Hence, observation was made herein of the microstructures of the composites by scanning electron microscopy (SEM) in order to investigate the morphology of the samples and, especially, the influence of the filler. The SEM micrograph in Figure 1a shows the neat PP sample was homogeneous in structure. The degree of homogeneity and the internal structures of the composites are visible in Figure 1b–f. The filler TC was distributed evenly as non-aggregated inclusions in the plate-type shapes, interrupting the continuity of the structure of the neat PP polymer [31,32,33]. The cryo-fractured surface cracks stemmed from the low affinity of the hydrophobic PP matrix and the hydrophilic TC filler, resulting in poor adhesion between the two phases on the interface [34]. This exerted a crucial effect on mechanical, thermal, and other properties, as described later. No effect was discerned on the microstructure of the resultant composites through differing the amount of EO supplemented into the TC structure at the given concentrations. This finding is not in accordance with other authors, who report that neat linseed or thyme EO, which had been directly mixed with the polymer matrix, caused a porous surface in films through them evaporating from the structure of the polymer [35,36]. As a consequence, the homogeneous structures described herein without such porous surfaces resulted from immobilizing the EO in the carrier, therefore, no free droplets formed in any of the matrices.

#### 3.1.2. Tensile Properties

Tensile testing was conducted to determine the effect of the TC and the incorporation of the EO on the mechanical properties of the prepared films. Figure 2 details Young’s modulus, tensile strength, and elongation at break of the neat PP and the PP/TC composites containing different amounts of immobilized caraway EO in their structure. The values obtained for tensile strength and elongation at break for neat PP (34.4 ± 2.4 MPa and 118 ± 5%, respectively) were in accordance with the technical data sheet on the polymer and the findings of other authors who have discussed the same type of homopolymer [37,38]. The TC has the capability of improving the rigidity and the strength of the PP matrix through lending it some reinforcement [39]. Figure 2 details that the Young’s modulus for PP/TC showed the most significant increase of up to 1634 ± 152 MPa as well as decrease in tensile strength (29.6 ± 1.7 MPa) and elongation (7.1 ± 0.7%). Similar results were reported by Karrad et al. [40] in a study on an HDPE/PS/TC system and the effects of TC on mechanical behaviour, wherein modulus values gradually increased in parallel with greater content of talc. Other studies confirm a comparable trend of the growth of modulus and the reduction of elongation values depending on the incorporation of talc into the PP matrix [32].

It was observed that adding the caraway EO reduced values for Young’s modulus by about 14%, 18%, 62%, and 73% and those for tensile strength by approximately 7%, 14%, 32%, and 63%, in line with rise in EO concentration. This behaviour could have been caused by the plasticizing effects of the EO (in free form caused by evaporation) in the polymer matrix, thereby heightening the ductile properties. Three theories exist to explain this plasticizing mechanism: (i) the lubricity theory, which considers that the plasticizer acts as a lubricant to reduce friction and facilitates mobility of the polymer chains to pass one another, thereby diminishing deformation; (ii) the gel theory, suggesting that a plasticizer disrupts and replaces polymer–polymer interactions that hold polymer chains together; and (iii) the free volume theory, whereby any polymeric material is defined as the internal space available in a polymer for the movement of chains [41,42,43]. However, a significant reduction was visible in the elongation at break of the filled samples in comparison with neat PP by about 30%, even in the one with the highest concentration of EO. As reported by Wu et al. [44], thymol EO functioned as a plasticizer, minimizing the intermolecular forces of the polymer chains, thus enhancing the flexibility and the extensibility of the films. Similar results were described by Llana-Ruiz-Cabello [23] in a study on PP films supplemented with oregano EO and allium extract intended for the packaging of meat products. This highlights that EOs not only provide an antibacterial function but also heighten the flexibility of PP composites which contain reinforcing fillers [28]. The improved mechanical properties can be also caused by the TC surface modification with EO considering that the modification of TC surface with EO presumably increases TC surface hydrophobicity and thus improves the interaction strength at the PP/TC interface.

### 3.2. Thermal Properties

#### 3.2.1. Thermogravimetric Analysis

Figure 3 displays thermograms obtained from thermogravimetric analysis (TGA) of neat PP, caraway EO, carrier TC with immobilized EO, and polymeric PP film filled with the TC/EO at the theoretical concentration of 20 wt% (PP/TC/20EO). The degradation temperature of the caraway EO, in approximate terms, equalled 140 °C, in agreement with data in the literature [11], and one decomposition peak was discerned related to complete destruction of organic matter. The thermal stability of the neat EO was slightly compromised after having been incorporated into the TC carrier due to access by heat facilitated through disruption caused by the porous structure of the filler aggregates. Inclusion of the enhanced filler into the polymer matrix, however, strengthened the thermal response of the composite. Thermal degradation of PP tends to be reported as occurring via random scission followed by a radical transfer process, with single-step thermal decomposition happening at around 500 °C [27,45]. The PP/TC/EO composites show two stages of degradation, though; the first was between 100 and 150 °C, due to a dehydration process where moisture was eliminated from the PP surface and the caraway EO evaporated; the other involved further degradation of the PP to its final products—propylene, pentane, 2-Methyl-1-pentene, and 2,4-Dimethyl-1-heptene. Hence, the degradation temperature of the PP/TC/20EO sample was observed at 443 °C, in comparison with 463 °C for the neat PP film, which indicates filling the PP matrix with a TC carrier is a suitable way to improve its thermal stability [46,47,48,49,50].

#### 3.2.2. Differential Scanning Calorimetry

The results given in Table 2 cover melting and crystallization temperatures and enthalpies of the film samples, as determined by differential scanning calorimetry and calculation of the degree of crystallization in per cent. Adding the TC into the PP matrix led neither to a shift in T_m_ (164.4 and 165.4 °C, see Figure 4) nor a change in temperature of T_c_, in agreement with research by others [26,51,52]. Nevertheless, TC had an impact on enthalpy, causing a decrease of about 15 J/g (see Table 2). The rise in crystallinity indicates TC is a good nucleating agent, since the PP has the potential to undergo transcrystallization on the TC surface. Such a trend was observed in a study by Ammar et al. [26], in which TC was incorporated as a reinforcing filler in a PP matrix. Note that slight changes in the T_m_ for each sample relate to an error in measurement common for this method [53]. As the EO content increased, the intensity of the melting peak diminished, indicating reduction in enthalpy; moreover, the intensity of the melting peak reduced by decreasing of PP content in the PP/TC/EO composites as well. This revealed that the crystallinity phase receded alongside increase in the content of the caraway EO in composites in comparison with PP/TC, as described in Table 2. While this demonstrated the effect of the EO on the crystallinity of the sample, the differences in T_m_ for the composite were negligible and thus were dismissed as unimportant; hence, the melting temperature was considered identical to the temperature interval for neat PP. The glass transition temperature (T_g_) was not observed at all due to its very low expression in the gathered data; notably, other authors have not given values for T_g_ either [7,26,27]. The crystallinity index (*X_C_*) was affected by the amount of EO and thus acted in dependence on concentration. Immobilizing the EO in the sample caused decline in values for *X_C_* in confirmation of the plasticizing effect and the TC surface hydrophobicity modification by EO incorporation, in comparison with PP and PP/TC films. The unexpected sharp decline in crystallinity in the sample with the highest EO concentration (20%) could have been due to the loading capacity of the TC as a carrier as per its interaction with the EO. Therefore, the decrease in crystallinity might have arisen through interactions between the polymer matrix and the free volume of the molecules of the caraway EO in the PP macromolecular network. A similar effect has been reported for PP, where carvacrol and thymol were directly added into a polymer matrix [54].

### 3.3. Qualitative Analysis of the Caraway EO

Identifying the target compounds in the caraway EO involved measuring the molecular ions present by gas chromatography on apparatus coupled to a single quadrupole mass spectrometer; the results are given in Table 3 and Figure 5. The yield of identified components equated to 98.9%, with 28 compounds being identified in total. Carvone was the major component in the EO derived from caraway seeds (66.5%), followed by limonene (23.2%). Numerous minor substances were also recognized, as listed in the literature, e.g., myrcene, linalool oxide and limonene oxide, linalool, and carveol [11,55,56,57,58]. Most EOs rich in carvone have proven very effective against a wide spectrum of human pathogenic fungi and bacteria. Carvone (2-Methyl-5-(prop-1-en-2-yl)cyclohex-2-en-1-one) is probably one of the most widespread monoterpenoids of natural compounds, owing to its biological and pharmacological properties, including anti-inflammatory, antimicrobial, antioxidant, antitumour, and insecticidal activities [7,10]. The results of this study concur with those in the literature, confirming the prevalence of carvone and limonene in caraway EO [55,58,59].

### 3.4. Release Kinetics of the Caraway EO

The release kinetics of the EO derived from *Carum carvi* L. seeds were performed by measurement that discerned the concentration of carvone as a major compound by Pyrolysis-GC/MS as well as calculation of the chromatographic area and further reference of the given numbers to the concentration of the compound in the samples. In order to further elaborate the release mechanism of the EO from the composites, the release profiles were fitted with an orthogonal polynomial fitting model. The coefficient of determination in the range 0.957–0.999 in all cases indicated predictions of regression were in very good agreement with the data.

Figure 6 reveals the significant influence exerted by the concentration of caraway EO in the samples on the rate of release, with considerable differences visible between the four types of films. All the samples exhibited a burst release within the first few hours of the experiment, which then plateaued due to the progressive depletion of carvone [7]. This is why those with higher concentrations of the EO demonstrated rapid and complete release of the EO entrapped (with a release rate of 90–100%). A lesser amount of EO brought about reduced and slower release of the compound, the actual release rate barely reaching 75%. The rate of the burst effect was dependent on concentration of EO, e.g., samples with greater concentrations of the EO showed higher rates; thus, larger amounts of the EO failed to incorporate in the TC aggregates and were able to evaporate. This finding is valid for samples containing high levels of the EO (PP/TC/10EO and PP/TC/20EO).

Release of the EO from samples with lesser content of the oil (PP/TC/3EO and PP/TC/7EO) happened in two stages, following the trend of substances immobilized in non-porous aggregated carriers such as TC. When an incorporated EO showed initial rapid release from an aggregated carrier with a surface more accessible to the release medium, which was also well adhered to the surface of the carrier, molecules inside the aggregates were freed by diffusion at a later time. The extent of release was incomplete, reaching 80% maximally in both samples, this potentially stemming from the lesser capability of the caraway EO to evaporate due to encapsulation in the carrier and, thus, the polymer matrix as well. The data obtained correspond with other studies [42,60,61].

In order to understand the processes behind the current results, despite the high level of research which has already taken place on various EOs as the antimicrobial agents of polymers, systematic studies on the principles of release of EO components from PP films are very scarce. There are diffusion and partitioning processes to be explored as major factors in regulating the freeing of the EO from the PP matrix, for example. It is worth noting that, in all cases investigated, the final amount of the EO released at equilibrium was lower than the nominal content in the films. This aspect can be ascribed to two phenomena: (a) the EO tends to volatilize, especially at the high temperatures required for melt processing; (b) a certain aliquot of the EO remains entrapped inside the matrix [7].

### 3.5. Study of Antimicrobial Properties

#### 3.5.1. Disk Diffusion Method

The inhibition zones of the EO derived from *Carum Carvi* L. and its diluted solutions in ethanol are given in Table 4. The EO generated larger zones of inhibition against the gram-positive bacterial strain (about 14 mm) compared to the gram-negative one (4.3 mm). Similar results were published in a study on the antimicrobial activity of bioactive components of essential oils derived from spices from Pakistan against salmonella and other multi-drug resistant bacteria [62]. According to research by Trombetta et al. [63], gram-negative bacteria are more resistant to EOs than gram-positive bacteria, a phenomenon caused by the differences between them in cell wall structure. As reported by Nazzaro et al. [64], the structure of the gram-positive bacteria cell wall allows hydrophobic molecules to move easily through the porins of the cells. In other words, gram-negative bacteria are less sensitive to the EOs because of their lipopolysaccharide outer membrane, which restricts the diffusion of hydrophobic compounds [65]. No inhibition zones were discerned for any of the film samples, possibly caused by the EO being well consolidated on the TC. Other authors have failed to demonstrate this advantage of incorporating an EO in the inert carrier in their studies on directly mixing EOs with different polymers; therein, the EOs migrated very fast and created zones of inhibition [42,60,66].

It can be concluded that the antimicrobial activity of the caraway EO under investigation is potentially caused by penetration of the oil into lipid assemblies and by consequent perturbation of the lipid fraction of the plasma membranes of the cells.

#### 3.5.2. Measurement of Antibacterial Activity on the Plastic Surfaces

The data on antibacterial activity on the surfaces of the samples are summarized in Table 5. Assays to test such activity against gram-positive and gram-negative bacteria by the prepared films with the EO showed it took place against both bacterial strains [67]. The mixture of PP and TC showed no antibacterial activity in either case; indeed, it is known from Kneuer et al. [68] that TC is a biologically inert and inactive material. The data obtained herein demonstrate the high antimicrobial efficiency of the EO at a theoretical concentration of more than 10% (actual EO content 4.9 ± 0.2 wt%). Comparing the results of antibacterial activity with those for release reveals that samples PP/TC/10EO and PP/TC/20EO lost in excess of 50% of the EO from the matrix during the first 24 h. Therefore, this amount of EO was freely available for interaction with the bacterial cells. It can be also seen that the caraway EO was more effective against the gram-positive bacterial strain, in accordance with the findings above. The antibacterial activity exerted by sample PP/TC/20EO was determined again after 5 weeks, following storage in a polyethylene bag at room temperature and the relative humidity of 40%. The results indicated that the sample still showed very strong antibacterial activity against both bacterial strains (*S. aureus* R ≥ 5.2, *E. coli* R ≥ 6.3).

## 4. Conclusions

The authors prepared antimicrobial polymer films based on polypropylene modified by caraway essential oil derived from *Carum carvi* L. seeds, which was immobilized on the carrier of talc, according to a thermoplastic process. The resultant films were compression moulded to an approximate thickness of 200 µm, which possessed a typical light grey-green hue and had the aroma of caraway. The effect of incorporating the caraway essential oil in the composites on the properties of the films were examined by morphological study, tensile testing, thermal analysis (TGA and DSC), gas chromatography, and antimicrobial evaluation.

Scanning electron microscopy of the cryo-fractured surfaces of the composites with the talc filler showed even distribution in the form of non-aggregated inclusions in the plate-type shapes, which disturbed the continuous structure of the neat polypropylene matrix. However, in all cases of caraway essential oil addition into the composition, it had no impact on the distribution of talc in polymer matter. Moreover, polypropylene/talc/essential oil composites showed comparable homogeneity in the entire volume of the samples. Tensile testing revealed that adding the essential oil reduced the tensile strength of the films but increased their elongation at break of as a consequence of exerting a plasticizing effect. Modifying the polypropylene with the caraway essential oil immobilized on a solid carrier of talc appropriately improved the thermal stability of the antimicrobial films. It was demonstrated that talc particles have an influence on crystallinity in polypropylene composite; talc already acted as a nucleating agent for polymer matrix. As a consequence, the release kinetics of caraway essential oil can be controlled depending on the input concentration of the volatile substance. The caraway essential oil, comprising carvone and limonene as its major compounds, generated larger zones of inhibition against gram-positive *Staphylococcus aureus* than gram-negative *Escherichia coli*. In this context, the prepared films showed antimicrobial activity against the pathogens of gram-positive *Staphylococcus aureus* and gram-negative *Escherichia coli* at the concentration of caraway essential oil of 4.9 ± 0.2 wt% and higher. Therefore, at a time of a global pandemic caused by the COVID-19 virus, materials such as the prepared PP/TC/EO films appear to be suitable for real-world applications that require antimicrobial action, including the packaging of food.

## Figures and Tables

**Figure 1 polymers-13-00906-f001:**
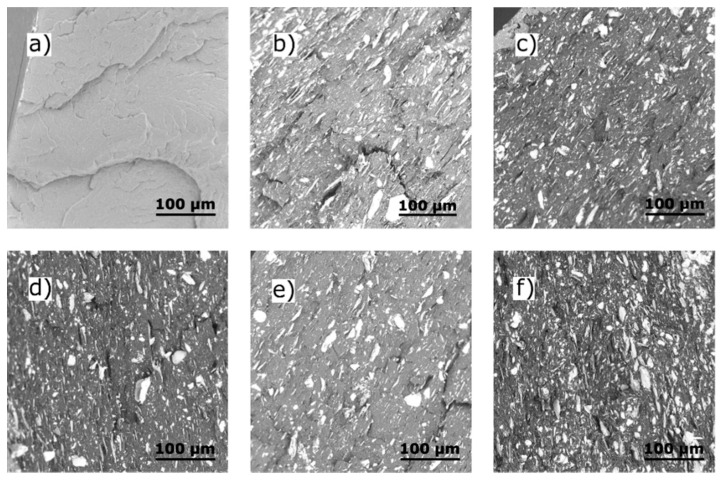
Scanning electron microscopy (SEM) micrographs of the fractured surfaces of (**a**) neat PP, (**b**) PP/TC, (**c**) PP/TC/3EO, (**d**) PP/TC/7EO, (**e**) PP/TC/10EO, and (**f**) PP/TC/20EO; magnification 700×.

**Figure 2 polymers-13-00906-f002:**
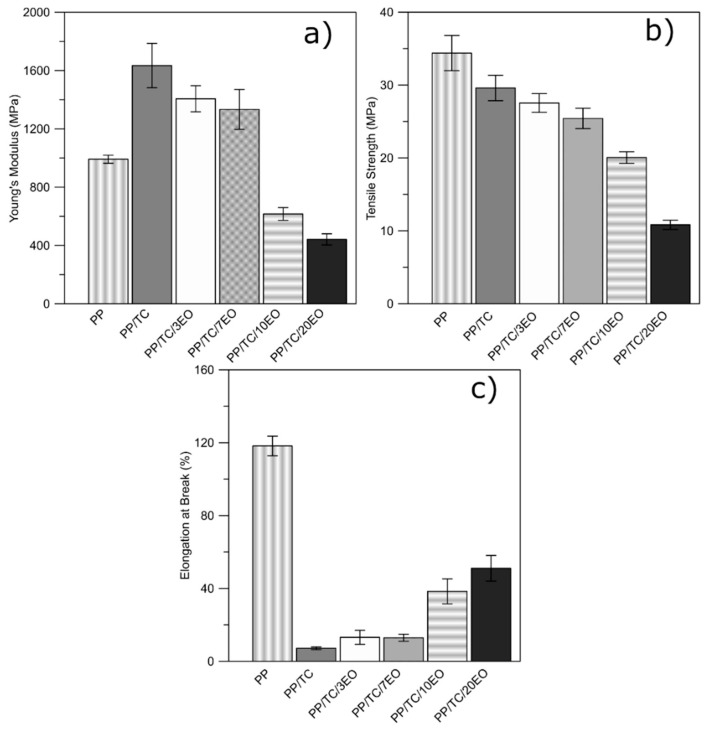
Young’s modulus (**a**), tensile strength (**b**), and elongation at break (**c**) of the neat PP and its composites with immobilized EO on the TC carrier.

**Figure 3 polymers-13-00906-f003:**
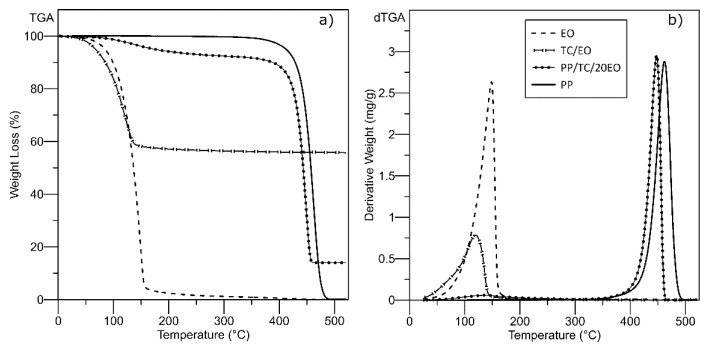
Thermogravimetric analysis of neat PP, caraway EO, TC/EO, and PP/TC/20EO film (**a**) TGA and (**b**) dTGA curves.

**Figure 4 polymers-13-00906-f004:**
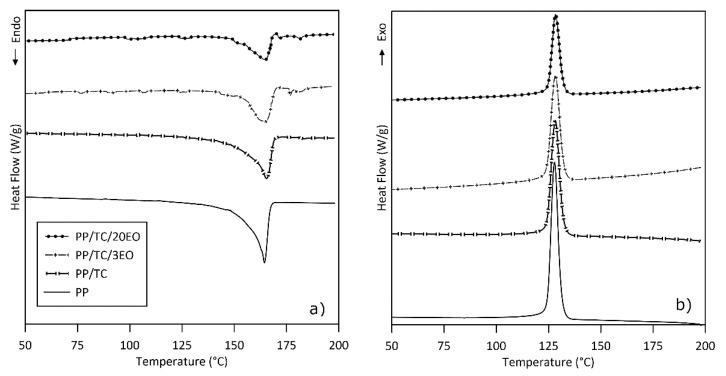
Differential scanning calorimetry readings (**a**) endothermic and (**b**) exothermic heat flow for the neat PP, PP/TC, PP/TC/3EO, and PP/TC/20EO films.

**Figure 5 polymers-13-00906-f005:**
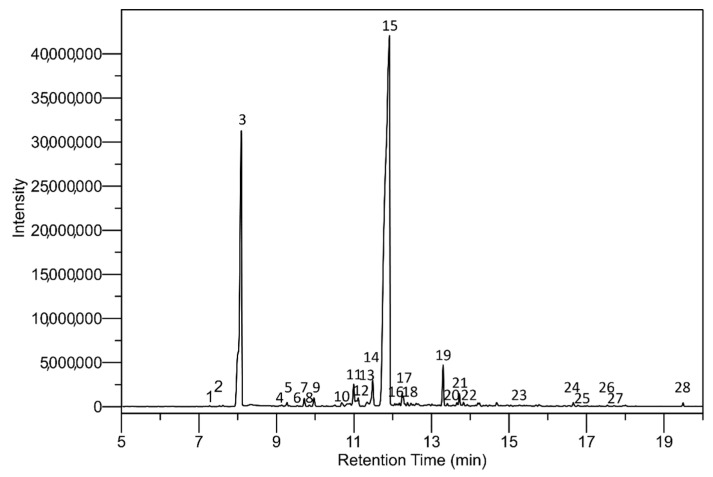
GC/MS chromatogram of caraway EO derived from *Carum carvi* L. seeds.

**Figure 6 polymers-13-00906-f006:**
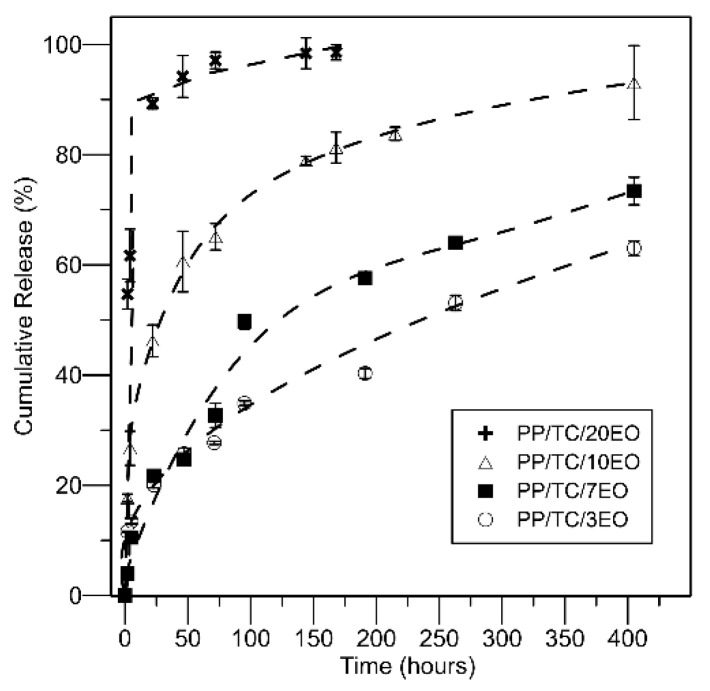
Cumulative release kinetics of the caraway EO from the films (40 °C; 50% relative humidity).

**Table 1 polymers-13-00906-t001:** Designation and composition of the samples.

Designation	PP (wt%)	TC (wt%)	Theoretical EO Content (wt%)	Actual EO Content (wt%) ^1^
PP	100	-	-	-
PP/TC	70	30	-	-
PP/TC/3EO	67	30	3	0.9 ± 0.1
PP/TC/7EO	63	30	7	1.7 ± 0.2
PP/TC/10EO	60	30	10	4.9 ± 0.2
PP/TC/20EO	40	30	20	7.1 ± 1.2

^1^ Determined by GC/MS. PP: polypropylene. TC: talc; EO: essential oil.

**Table 2 polymers-13-00906-t002:** Differential scanning calorimetry (DSC) results for all the investigated films.

Sample	T_m_ (°C)	T_c_ (°C)	∆*H_f_* (J/g)	*X_C_* (%)
PP	164.4	127.9	−98.3	47.0
PP/TC	165.4	128.1	−83,8	57.3
PP/TC/3EO	164.9	128.4	−59.3	41.0
PP/TC/7EO	164.9	128.7	−60.4	42.3
PP/TC/10EO	164.5	127.3	−63.6	46.6
PP/TC/20EO	164.8	128.5	−44.9	34.1

**Table 3 polymers-13-00906-t003:** Relative values for the components in the EO (as a percentage of the total peak area) derived from *Carum carvi* L. seeds.

Peak #	Retention Time (min)	Retention Index	CAS #	Name	M_w_	Area (%)
1	7.274	958	123-35-3	beta-Myrcene	136	ms ^1^
2	7.528	1005	124-13-0	Octanal	128	ms
3	8.092	1018	5989-27-5	d-Limonene	136	23.2
4	9.132	1073	1195-32-0	*p*-Cymenene	132	ms
5	9.271	1082	78-70-6	Linalool	154	0.2
6	9.538	1206	99-48-9	Carveol	152	ms
7	9.716	1136	1845-30-3	*cis*-Verbenol	152	0.4
8	9.847	1088	6090-09-1	Limona ketone	138	0.1
9	9.966	1120	22771-44-4	*p*-Mentha-2,8-dienol	152	0.5
10	10.683	1174	141-27-5	Citral	152	0.2
11	10.993	1179	5948-04-9	*trans*-Dihydrocarvone	152	1.3
12	11.108	1179	3792-53-8	*cis*-Dihydrocarvone	152	0.5
13	11.340	1031	6909-30-4	*trans*-Limonene oxide	152	0.4
14	11.483	1206	1197-07-5	*trans*-Carveol	152	1.7
15	11.913	1190	2244-16-8	d-Carvone	150	66.5
16	12.056	1202	39903-97-4	*trans-*Carvone oxide	166	0.1
17	12.243	1207	2111-75-3	Perilla aldehyde	150	0.9
18	12.381	1335	57576-09-7	Isopulegol acetate	196	0.1
19	13.300	1031	1195-92-2	*cis*-Limonene 1,2-epoxide	152	2.0
20	13.409	1346	1946-00-5	Limonene-1,2-diol	170	0.1
21	13.655	1190	104-46-1	Anethole	148	0.2
22	13.826	1097	78-59-1	Isophorone	138	0.1
23	15.267	1278	499-75-2	Carvacrol	150	0.1
24	16.660	1507	1139-30-6	Caryophyllene oxide	220	0.2
25	16.780	1266	97-41-6	Ethyl chrysanthemate	196	0.1
26	17.540	1386	489-39-4	Alloaromadendrene	204	0.1
27	17.710	1507	1139-30-6	Caryophyllene oxide	220	ms
28	19.491	1754	502-69-2	2-Pentadecanone, 6,10,14-trimethyl-	268	0.1

^1^ ms—Detected substance in minor concentration (less than 0.1%). The components are listed in order of elution in the polar column.

**Table 4 polymers-13-00906-t004:** Inhibition zones of the caraway EO from *Carum carvi* L. seeds.

	Concentration (vol%)	Inhibition Zone Diameter (mm)	
		*Staphylococcus aureus*	*Escherichia coli*
EO from *Carum carvi* L.	0.1	0	0
	1.0	0	0
	10	6.4 ± 0.7	1.3 ± 0.1
	100	14.0 ± 2.3	4.3 ± 0.6

**Table 5 polymers-13-00906-t005:** Antibacterial activity (R) of the prepared films with the caraway EO against gram-positive and gram-negative bacterial strains at ISO 22196:2011 standard.

Sample	Antibacterial Activity (R)	
	*Staphylococcus aureus*	*Escherichia coli*
PP/TC	0	0
PP/TC/3EO	0.7	0
PP/TC/7EO	1.4	0
PP/TC/10EO	≥5.3	≥6.4
PP/TC/20EO	≥5.3	≥6.4

## Data Availability

The data presented in this study are available on request from the corresponding author.

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
