# Peer review of "Immobilization of Caraway Essential Oil in a Polypropylene Matrix for Antimicrobial Modification of a Polymeric Surface"

_polymers, 2021, doi:10.3390/polym13060906_

Round 1
Reviewer 1 Report
Comments
- Table 1 - to explain, how was the actual EO content determined in samples? Why is not discussed the actual but theoretical EO content in the Results and Discussion?
- page 7 - would it not be possible to find in the references a comparison of mechanical properties with PP/TC instead of HDPE/PS/ TC?
- age 9 -„Nevertheless, TC had an impact on enthalpy, causing a decrease of about 15°C (see Figure 4).“ – This statement based on Fig. 4 is not true. It needs to be corrected.
- page 9 - „As the EO content increased, the intensity of the melting peak diminished, indicating reduction in enthalpy.“ – The intensity of the melting peak diminished by decreasing of PP content in the PP/TC/EO composites, too. Measured „the enthalpy“ is not relevant value for comparison of samples without calculation of crystalinity only on the intensity of peak on the figure.
- page 9 - „This revealed that the crystallinity phase receded alongside increase in the content of the caraway EO, as described in Table 2.“ - Is the reduction of PP crystallinity of PP/TC/EO samples compared with crystallinity of near PP or PP/TC composite? It is not the same. It needs to be explained better.
Author Response
Response to Reviewer 1 Comments
Point 1: Table 1 - to explain, how was the actual EO content determined in samples? Why is not discussed the actual but theoretical EO content in the Results and Discussion?
Response 1: The actual EO content was evaluated following the procedure reported for the release study and is described in section 2.6. The discussion related to the actual content is reported in the Results and Discussion section. The samples were named by the theoretical EO content for better legibility of sample labels.
Point 2: page 7 - would it not be possible to find in the references a comparison of mechanical properties with PP/TC instead of HDPE/PS/ TC?
Response 2: We add a comparison with the study about the influence of processing and particle morphology on the final properties of polypropylene/talc nanocomposites.
Point 3: page 9 -„Nevertheless, TC had an impact on enthalpy, causing a decrease of about 15°C (see Figure 4).“ – This statement based on Fig. 4 is not true. It needs to be corrected.
Response 3: The statement has been changed.
Point 4: page 9 - „As the EO content increased, the intensity of the melting peak diminished, indicating reduction in enthalpy.“ – The intensity of the melting peak diminished by decreasing of PP content in the PP/TC/EO composites, too. Measured „the enthalpy“ is not relevant value for comparison of samples without calculation of crystalinity only on the intensity of peak on the figure.
Response 4: We went through the raw data again and the statement has been improved.
Point 5: page 9 - „This revealed that the crystallinity phase receded alongside increase in the content of the caraway EO, as described in Table 2.“ - Is the reduction of PP crystallinity of PP/TC/EO samples compared with crystallinity of near PP or PP/TC composite? It is not the same. It needs to be explained better.
Response 5: A more detailed explanation has been reported.
Reviewer 2 Report
The manuscript “Immobilization of caraway essential oil in a polypropylene matrix for antimicrobial modification of a polymeric surface” deals with the production of antibacterial polypropylene composites using caraway essential oil. Several analyses were performed on these composites that showed a good activity against Staphylococcus aureus and Escherichia coli. The work is well organized and relevant results were obtained. Therefore, the publication is recommended; but after some revisions.
In particular:
- Introduction. The state of art about the use of natural origin active compounds in biocompatible polymers can be enlarged; see, for instance, the works of Cardea et al., Comparative study of PVDF-HFP-curcumin porous structures produced by supercritical assisted processes, Journal of Supercritical Fluids, 2018, 133, pp. 270-277; Ibrahim et al., Synergistic antimicrobial effect of xylitol with curcumin: Water vapor barrier, mechanical and thermal properties of PSS/PVA packaging films, International Journal of Applied Engineering Research, 2017, 12, pp. 10360-10366.
- Results and discussion. SEM images reported in Figure 1 are not meaningful because are too small and the magnification is not indicated. Please, change it.
- Conclusions. In the present form this paragraph is a summary of the results obtained. Please, rewrite in a more critical way.
Author Response
Response to Reviewer 1 Comments
Point 1: Introduction. The state of art about the use of natural origin active compounds in biocompatible polymers can be enlarged; see, for instance, the works of Cardea et al., Comparative study of PVDF-HFP-curcumin porous structures produced by supercritical assisted processes, Journal of Supercritical Fluids, 2018, 133, pp. 270-277; Ibrahim et al., Synergistic antimicrobial effect of xylitol with curcumin: Water vapor barrier, mechanical and thermal properties of PSS/PVA packaging films, International Journal of Applied Engineering Research, 2017, 12, pp. 10360-10366.
Response 1: The presented work aims to incorporate essential oil into a composite material by thermoplastic processing. We went through the manuscript suggested and we decided to do not add as an additional reference as it does not fit with the aim of our work. We have designed the introduction part on the basis of the available literature, which is closely related to antimicrobial additives based on essential oils mostly for food packaging.
Point 2: Results and discussion. SEM images reported in Figure 1 are not meaningful because are too small and the magnification is not indicated. Please, change it.
Response 2: Magnification has been added in the figure caption. Unfortunately, due to government restrictions on the covid-19 pandemic, we are not able to perform a new SEM evaluation in a shorter time. However, the essential structural information about the material is illustrated in Figure 1. The structure in the entire volume of the composition is very interesting from a technological / processing point of view.
Point 3: Conclusions. In the present form this paragraph is a summary of the results obtained. Please, rewrite in a more critical way.
Response 3: The conclusion has been rewritten.
Reviewer 3 Report
The manuscript Polymers-1125902 reports the preparation of polypropylene (PP) matrices containing caraway essential oil (EO)-immobilized talc (TC) to obtain antimicrobial packaging. Different amounts of EO were entrapped in the matrices to investigate the effect of such parameter on the physical and biological properties of the matrices themselves.
General comment
The paper is clearly presented and the topic is worth of investigation. The approach used to entrap EO in PP is interesting because can minimize the EO evaporation issue found in different entrapping methods. I have just few suggestions that the authors can consider to revise their manuscript. Particularly, I would like the authors to consider a different hypothesis, rather than the plasticizing effect, to explain the effect of EO content on PP mechanical and thermal properties (read below). Also the presentation of antibacterial activity data needs revision.
Specific comments
1) Please define better At in the equation 3. Actually, it is the first time I see this equation to express the antibacterial activity of polymer surfaces. I wonder which are the lower and upper limits (No activity – highest activity). Is there any reference to support the use of such equation?
2) Figure 2. I would comment the mechanical properties of the matrices containing increasing amount of EO considering that the modification of Talc surface with EO presumably increases TC surface hydrophobicity and thus improves the interaction strength at the polymer/TC interface (higher adhesion). Such better surface compatibility between PP and the filler may justify the absence of cracks in the EO containing PP/TC composites compared to the neat PP/TC composite, observed by SEM. Of course, such better PP/TC adhesion favors the PP ductile break over the brittle rupture, with a consequent increase in the elongation a break with increasing EO content. I don’t agree with the hypothesis of a plasticizing effect of EO because EO is not mixed with the polymer (essential for a molecule to exert a plasticizing effect) but anchored on the TC surface.
3) Table 2. Please consider the above suggestion also for the discussion of the decrease in crystallinity of PP with increasing EO content.
4) Table 5. What does an activity higher than or equal to 5.3 (or 6.4) mean? Please, either justify the use of R or change the way to present the antibacterial activity data.
Author Response
Response to Reviewer 3 Comments
Point 1: Please define better At in the equation 3. Actually, it is the first time I see this equation to express the antibacterial activity of polymer surfaces. I wonder which are the lower and upper limits (No activity – highest activity). Is there any reference to support the use of such equation?
Response 1: The definition of the parameter At has been improved.
The expression of the results is made exactly according to the ISO 22196:2011 standard (see reference number [30]), in which the antibacterial activity is calculated as a difference between the logarithm of the viable cell count in the treated sample surface and the untreated sample surface after inoculation and incubation with the bacteria.
The reference to support the methodology, as well as the equation (2) and (3), is mentioned in section 2.7.2.
Point 2: Figure 2. I would comment the mechanical properties of the matrices containing increasing amount of EO considering that the modification of Talc surface with EO presumably increases TC surface hydrophobicity and thus improves the interaction strength at the polymer/TC interface (higher adhesion). Such better surface compatibility between PP and the filler may justify the absence of cracks in the EO containing PP/TC composites compared to the neat PP/TC composite, observed by SEM. Of course, such better PP/TC adhesion favors the PP ductile break over the brittle rupture, with a consequent increase in the elongation a break with increasing EO content. I don’t agree with the hypothesis of a plasticizing effect of EO because EO is not mixed with the polymer (essential for a molecule to exert a plasticizing effect) but anchored on the TC surface.
Response 2: We appreciate the comment and it has been added in the Results and Discussion section.
We agree that EO is not mixed with the polymer directly, but from the release kinetic trend is observable that the EO slowly evaporate (EO is not covalently bonded on TC surface). It suggests that a certain amount of EO interacts with the polymer matrix.
Point 3: Table 2. Please consider the above suggestion also for the discussion of the decrease in crystallinity of PP with increasing EO content.
Response 3: The changes in crystallinity have been explained.
Point 4: Table 5. What does an activity higher than or equal to 5.3 (or 6.4) mean? Please, either justify the use of R or change the way to present the antibacterial activity data.
Response 4: The results in the Table 5 are presented according to the ISO standard. For this reason, we prefer to maintain the current expression of the results. Moreover, the reported expression of R is widely used and accepted.
An activity higher than or equal to 5.3 or 6.4 (Ut value of references) means the average of the common logarithm of the viable bacteria (in cells/cm2), recovered from the reference specimens after 24 h. In case that we will prepared bacterial suspension with higher cells concentration it can happened that the number of logarithm will be higher. This designation means that no visible bacteria were observed on the test sample surface after 24 h incubation.
Round 2
Reviewer 3 Report
The authors have addressed my suggestions.